Similar acute physiological responses from effort and duration matched leg press and recumbent cycling tasks

Steele James james.steele@solent.ac.uk 1 2
Butler Andrew 1
Comerford Zoe 1
Dyer Jason 1
Lloyd Nathan 1
Ward Joshua 1
Fisher James 1
Gentil Paulo 3
Scott Christopher 4
Ozaki Hayao 5
1 School of Sport, Health, and Social Sciences, Southampton Solent University , United Kingdom
2 ukactive Research Institute, ukactive , London , United Kingdom
3 Faculty of Physical Education, Federal University of Goiás , Brazil
4 Department of Exercise, Health, and Sport Sciences, University of Southern Maine , United States of America
5 Graduate School of Health and Sports Science, Jutendo University , Japan
Ramírez-Campillo Rodrigo
Electronic publication date: 2018 Feb 28
Publication date: 2018
Volume: 6
Electronic Location ID: e4403
Received 2017 Dec 2; Accepted 2018 Feb 1
Copyright: ©2018 Steele et al.
Copyright year: 2018
Copyright holder: Steele et al.
License: This is an open access article distributed under the terms of the Creative Commons Attribution License, which permits unrestricted use, distribution, reproduction and adaptation in any medium and for any purpose provided that it is properly attributed. For attribution, the original author(s), title, publication source (PeerJ) and either DOI or URL of the article must be cited.
License URL: https://creativecommons.org/licenses/by/4.0/

Keywords: Electromyography, Lactate, Energy expenditure, Muscle swelling, Exercise modality

Funding: The authors received no funding for this work.

==============================
The present study examined the effects of exercise utilising traditional resistance training (leg press) or ‘cardio’ exercise (recumbent cycle ergometry) modalities upon acute physiological responses. Nine healthy males underwent a within session randomised crossover design where they completed both the leg press and recumbent cycle ergometer conditions. Conditions were approximately matched for effort and duration (leg press: 4 × 12RM using a 2 s concentric and 3 s eccentric repetition duration controlled with a metronome, thus each set lasted  60 s; recumbent cycle ergometer: 4 × 60 s bouts using a resistance level permitting 80–100 rpm but culminating with being unable to sustain the minimum cadence for the final 5–10 s). Measurements included VO2, respiratory exchange ratio (RER), blood lactate, energy expenditure, muscle swelling, and electromyography. Perceived effort was similar between conditions and thus both were well matched with respect to effort. There were no significant effects by ‘condition’ in any of the physiological responses examined (all p > 0.05). The present study shows that, when both effort and duration are matched, resistance training (leg press) and ‘cardio’ exercise (recumbent cycle ergometry) may produce largely similar responses in VO2, RER, blood lactate, energy expenditure, muscle swelling, and electromyography. It therefore seems reasonable to suggest that both may offer a similar stimulus to produce chronic physiological adaptations in outcomes such as cardiorespiratory fitness, strength, and hypertrophy. Future work should look to both replicate the study conducted here with respect to the same, and additional physiological measures, and rigorously test the comparative efficacy of effort and duration matched exercise of differing modalities with respect to chronic improvements in physiological fitness.

Introduction

Physical activity and exercise appears to offer protection against all-cause mortality (Nocon et al., 2008; Paffenbarge Jr et al., 1986) in a seemingly dose-response fashion (Byberg et al., 2009; Lee & Skerrett, 2001; Loprinzi, 2015a). Thus, public health guidelines currently recommend the accumulation of a minimum duration of physical activity and/or exercise per week (a combination of 30 min of moderate (50–70% max heart rate) five times per week and/or 20 min of vigorous (70–80% max heart rate) three times per week; Haskell et al., 2007). The efficacy of these recommendations could be considered disappointing in view of recent studies, showing that only a marginal reduction in all-cause mortality occurs when they are met (Lee et al., 2011; Wen et al., 2011; Loprinzi, 2015a). However, evidence is accumulating that outcome measures such as cardiorespiratory fitness (Lee et al., 2011; Wen et al., 2011; Kodama et al., 2009; Loprinzi, 2015b), strength (Buckner, Loenneke & Loprinzi, 2015; Leong et al., 2015; Loprinzi & Loenneke, 2016; Loprinzi, 2016; Newman et al., 2006; Ruiz et al., 2008; Strand et al., 2016), and muscle mass (Srikanthan & Karlamangla, 2014; Sriknathan, Horwich & Tseng, 2016) may be stronger predictors of health and longevity. Further, there is growing evidence of the potential efficacy of higher intensity of effort approaches, and so the paradigmatic shift in thinking towards an effort, rather than cumulative volume, driven model is beginning (Biddle & Batterham, 2015; Phillips & Winett, 2010; Steele et al., 2017a).

The primary focus of investigations regarding higher effort exercise approaches has been colloquially termed ‘cardio’ exercise modalities (i.e., locomotive and ambulatory modes such as cycling, running, rowing, incline walking, and stairclimbing; Biddle & Batterham, 2015; Phillips & Winett, 2010; Steele et al., 2017a). It has recently been argued, however, that resistance training (i.e., free weights, resistance machines, bodyweight/callisthenic exercises, resistance bands, etc.) may be a modality that also fits the higher effort paradigm as it is typically performed at a relatively high effort, for a relatively brief duration, and relatively infrequently (Steele et al., 2017a). Effort in resistance training has been defined as being determined primarily by proximity to momentary failure (i.e., when the demands of the exercise match or exceed the current ability to meet those demands; Steele et al., 2017b; Steele et al., 2017c; Steele, 2014). Phillips & Winett (2010) have noted previously with respect to resistance training that “...effort is internal to the person, can be created with a variety of protocols, and is not dependent upon a specific amount of external force.” Indeed, a more encompassing definition of effort is that it is the perception associated with attempting to achieve a particular demand, or set of demands, and which is determined by the current ability to meet the task demands relative to those demands. As such, this conceptualisation of effort could also be applied across exercise modalities (e.g., proximity to task failure in cycling) to better understand whether they impart similar or differential physiological effects. Historically, resistance training modalities and ‘cardio’ training modalities have been dichotomised with respect to the physiological stimulus they provide and chronic adaptations they produce (Nader, 2006). Indeed, signalling pathways responsible for the typical adaptations that seem to occur from either modality are apparently separate (Atherton et al., 2005). It has, however, been proposed that exercise modality may have little impact upon acute responses or adaptations assuming intensity of effort is similarly high (Fisher & Steele, 2014).

The adaptations in cardiorespiratory fitness typically thought to occur as a result of modalities of exercise such as cycling also have been reported to occur as a result of resistance training, though primarily if intensity of effort is sufficiently high (i.e., to momentary failure; Steele et al., 2012) and seemingly irrespective of the manipulation of other variables (i.e., load, set volume, rest periods, and frequency; Ozaki et al., 2013a). Conversely, the adaptations in strength and hypertrophy thought typically to occur from resistance training modalities of exercise have been found to occur as a result of ‘cardio’ modalities (Konopka & Harber, 2014), though again this seems primarily to be the case if they are performed with a high effort (i.e., combined with blood flow restriction, or with close proximity to failure such as interval training or sprinting; De Oliviera et al., 2016; Lundberg et al., 2013; Ozaki et al., 2015; Ozaki et al., 2013b). Despite this, studies directly comparing resistance training and ‘cardio’ training modalities upon these chronic adaptations contrast in their findings with some showing certain adaptations to be similar (Messier & Dill, 1985; Sawczyn et al., 2015; Hepple et al., 1997; Jubrias et al., 2001) and others showing some adaptations to differ (Hepple et al., 1997; Jubrias et al., 2001; Farup et al., 2012; Goldberg, Elliot & Kuehl, 1994; Wilkinson et al., 2008). These contrasting findings though may be due to the manner in which comparisons have been made. Indeed most comparisons have lacked parity in variables such as effort and duration. As noted, effort may be of importance and where this has been controlled between interventions (e.g., ‘high intensity interval training’ and high effort resistance training), recent work suggests that there may be little difference in the adaptations produced (Androulakis-Korakkakis et al., 2017; Álvarez et al., 2017).

Many of the physiological responses may be similar between exercise modes, assuming variables such as intensity of effort are controlled, and if this is the case, different modes may therefore result in similar adaptations. For example, it is thought that, similarly to ‘cardio’ modalities, resistance training to a high intensity of effort results in maximal aerobic and anaerobic metabolism, and as such local working muscle oxygen utilisation (VO2) and lactate production (Steele et al., 2012). Indeed, when matched for physiological effort (at the load/power eliciting individual ventilatory threshold/onset of blood lactate accumulation at 4 mmol L−1), resistance training modalities (half squats and bench press) have been shown to elicit similar acute VO2 to ‘cardio’ modality exercise (lower and upper body cycling (Vilaca-Alves et al., 2016). Further, high effort resistance training increases muscle water content (Giessing et al., 2016; Ribiero et al., 2014) resulting in muscle swelling that appears to be largely independent of external load (Loenneke et al., 2016). As such it is perhaps unsurprising that high effort aerobic modalities also increase muscle water content (Mora-Rodriguez et al., 2016) and result in muscle swelling (Ozaki et al., 2013b). In consideration of Henneman’s size principle, motor unit recruitment should also be similar between muscular actions when they are performed to a near maximal effort (Potvin & Fuglevand, 2017) and indeed this has been argued to be the case for resistance training whether performed at high or low loads (Fisher, Steele & Smith, 2017; Vigotsky et al., 2017). Thus, it might be expected that low force/load ‘cardio’ modalities might also produce high motor unit recruitment if performed with close proximity to momentary failure and thus high effort. However, in many studies of typical ‘cardio’ modes of exercise peak electromyographical amplitudes only achieve ∼10–50% of maximum voluntary contraction (Ericson et al., 1985; Marsh & Martin, 1995). Further, normalised electromyographic amplitudes appear to be greater during typical resistance training (single leg knee extension) compared with ‘cardio’ mode (single leg cycle ergometry) exercise performed to volitional failure (Noble et al., 2017). Although, time to task failure in Noble et al. (2017) was unclear and amplitude based analyses may not reflect the entirety of motor units recruited where task durations differ, particularly if differing recruitment patterns are occurring (i.e., sequential recruitment of low to high threshold during low force tasks, and simultaneous recruitment of both low and high threshold motor units during high force tasks; Fisher, Steele & Smith, 2017; Vigotsky et al., 2017; Potvin & Fuglevand, 2017; Enoka & Duchateau, 2015). However, where resistance training modes (knee extension) and ‘cardio’ modes (cycling) have been performed to momentary failure (thus controlling effort), and frequency based analyses applied, evidence suggests that similar recruitment of motor units may occur (Kuznetsov et al., 2011).

It seems that there may be greater similarity in the physiological responses to differing exercise modes than assumed under the typical historical dichotomisation of resistance training and ‘cardio’ training. Indeed the conceptualisation of exercise from the perspective of an effort based model (Steele et al., 2017a; Fisher & Steele, 2014) hypothesises that exercise modality may have little impact upon acute responses or adaptations assuming intensity of effort is high. The practical implications of this are considerable as it suggests that persons wishing to attain the greatest health and fitness benefits from engaging in exercise may be able to choose based upon personal preference from a wide range of possible modalities to produce a similar physiological stimulus, so long as intensity of effort is high. However, though studies independently report many responses to be similar, studies conducting direct comparisons of resistance training and ‘cardio’ exercise modalities using ecologically valid approaches (i.e., reflecting typical training approaches employed in real word settings) are lacking. Previous studies comparing acute responses to effort matched resistance training and ‘cardio’ exercise modalities have either not matched protocols employed for duration or used approaches that are not representative of typical exercise programs (Vilaca-Alves et al., 2016; Noble et al., 2017; Kuznetsov et al., 2011). Considering this, the aim of the present study was to examine the effects of training utilising traditional resistance training (leg press) or ‘cardio’ exercise (recumbent cycle ergometry) modalities whilst attempting to approximately control for effort and duration within ecologically valid protocols, upon acute physiological responses (VO2, respiratory exchange ratio [RER], blood lactate, energy expenditure, muscle swelling, and electromyography). It was hypothesised that, when effort and duration are matched, the two modalities would result in similar acute physiological responses.

Materials and Methods

Ethical approval

The study was approved by the Health, Exercise, and Sport Science ethics committee at Southampton Solent University (ID No. 316) and was conducted in accordance with the Declaration of Helsinki.

Study design

A within session randomised crossover design was adopted to examine the acute physiological effects of two different exercise modes; resistance training (leg press) and ‘cardio’ exercise (recumbent cycle ergometry). The within session design was used to reduce measurement error occurring across session’s (particularly for electromyography and ultrasound) and a randomised counterbalanced order used to mitigate any order effects and resultant fatigue. The two exercise modes were chosen to reflect the typical ecologically valid conceptualisation of what constitutes high effort ‘resistance training’ and what constitutes high effort ‘cardio exercise’. Attempts were made to approximately match the effort and duration of both conditions whilst reflecting ecologically valid high effort approaches to the performance of either mode. Thus, the resistance training mode involved performance of four sets of 12 repetition maximum (RM; using a 2 s concentric and 3 s eccentric duration—thus ∼60 s total per set), and the aerobic exercise mode involved 4 sets of sprints lasting 60 s in duration using a resistance meaning the participants could not sustain the required cadence (80 rpm) by the end of the set. Both conditions allowed 4 min rest between sets. VO2 and RER were examined during the exercise and rest periods, blood lactate was measured at the end of each exercise period, energy expenditure was estimated for both the exercise and rest periods, muscle swelling was measured pre and post both conditions, and electromyography was measured throughout the exercise periods.

Participants

Nine healthy male adult participants were recruited for the study. Participants were recreationally active but had not been involved in any structured exercise interventions for the previous six months. Exclusion criteria included; unstable angina, recent cardiac infarction, uncompensated heart failure, severe valvular illness, pulmonary disease, uncontrolled hypertension, kidney failure, orthopaedic/neurological limitations to exercise, cardiomyopathy, scheduled operations during the study period, current drug or alcohol abuse, or current participation in a parallel study. All participants were provided with a participant information sheet and gave written informed consent. Participant demographics are shown in Table 1.

Table 1 Participant demographics.

Variable	Mean (SD)	
Age (years)	26 (10)	
Height (cm)	179.7 (6.2)	
Body (kg)	87.2 (10.5)	
Body Mass Index (kg m2)	27.1 (3.5)	
Systolic blood pressure (mmHg)	129.6 (6.7)	
Diastolic blood pressure (mmHg)	74.2 (8.3)	

Equipment

Height was measured using a wall mounted stadiometer (Holtan ltd, Crymych, Dyfed), body mass measured using digital scales (Life Measurement Inc, Concord, CA, USA) and Body Mass Index (BMI) calculated. Resting blood pressure was measured whilst seated using an automated sphygmomanometer (Bosch and Sohn Germany, Jungingen, Germany). Expired gases were measured using a Viasys Oxycon Pro on-line gas analysis system (Jaeger, Bodnegg, Germany). Blood lactate measures were made using a Biosen C Line blood analyser (EKF Diagnostic, Germany). Muscle swelling was determined from measurements of muscle thickness using an M7 Diagnostic Ultrasound System (Shenzhen Mindray Bio-Medical Electronics Co. Ltd., Shenzen, China). Electromyography (EMG) was measured using a Trigno Digital Wireless EMG System (Delsys,USA). Perceived effort and discomfort were also recorded for each exercise condition using two separate 11 point scales to examine whether perceived effort was similar between conditions (Steele et al., 2017c). This was in order to avoid the cofounding effects of participants anchoring perceived effort upon perceived discomfort when traditional rating of perceived exertion scales are used without instruction to differentiate the two. The exercise conditions were performed using a Nautilus Nitro Evo Leg Press (Nautilus, Vancouver, USA) and a Cybex 530R Recumbent Cycle Ergometer (Cybex, Coalville, Leicestershire, UK).

Participant testing procedures

Participants were required to visit the laboratory for two testing sessions separated by 3–5 days. On the first session participants underwent testing in a randomised counterbalanced order to determine the leg press 12RM loads and recumbent cycle ergometer resistance levels to be used during the second testing session. A rest period of 20 min was permitted between each test. For 12RM determination participants first performed a set of 12 repetitions using a load of ∼50% their estimated 12RM. Participants were then subsequently permitted up to maximum of five attempts with 4 min of rest permitted between attempts to determine the 12RM load to be used for the leg press. Repetitions were performed using a using a 2 s concentric and 3 s eccentric repetition duration controlled with a metronome. Thus, the 12RM that was determined permitted ∼60 s in total of time under load. For determination of recumbent cycle ergometer resistance level participants first performed 60 s of cycling at the lowest resistance level at ∼60 rpm. Participants were then permitted up to a maximum of five attempts with 4 min of rest permitted between attempts to determine the resistance level (ranging 1 to 20) that would permit them to cycle for 60 s between 80–100 rpm culminating in failure to maintain a minimum cadence of 80 rpm over the final 5–10 s of the sprint.

On the second testing session participants underwent testing for acute physiological responses to both the leg press and recumbent cycle ergometer in a randomised counterbalanced order. Attempts were made to match the effort and duration of exercise performed in each condition and both the leg press and recumbent cycle were chosen in order to approximately match the musculature and positioning during each condition. A rest period of 20 min was permitted between each condition.

The leg press condition involved participants performing 4 sets of 12RM. Repetitions were performed using a 2 s concentric and 3 s eccentric repetition duration controlled with a metronome. Thus, each set lasted ∼60 s in total. If participants could not perform all 12 repetitions then the load was decreased for the next set by ∼5%. If participants were able to exceed all 12 repetitions then the load was increased for the next set by ∼5%. A rest of 4 min was permitted between each set.

The recumbent cycle ergometer condition involved participants performing 4 sets of sprints lasting 60 s in duration. The resistance level was set so that participants could cycle between 80–100 rpm but such that each set culminated in the participants being unable to sustain the minimum cadence. If participants could not sustain the required minimum cadence (80 rpm) prior to the last 5–10 s of the set then the resistance level was decreased by 1 for the next set. If participants still exceeded the required minimum cadence (80 rpm) over the last 5–10 s of the set then the resistance level was increased by 1 for the next set. A rest of 4 min was permitted between each set.

VO2, RER, blood lactate, and energy expenditure

Expired gases were collected throughout both the entirety of each condition, including both exercise and rest periods, and for a further 4 min rest post completion of the final set of either condition. Both mean and peak VO2 and RER were examined for each set, rest period, and also the averages across sets, rest periods, and the entire duration of each condition. Capillary blood samples were taken at rest and immediately post each set of exercise in each condition. This was performed using the procedures outlined by Maughan, Leiper & Greaves (2011). The tip of the participant’s middle finger on their non-dominant hand was cleaned using an alcohol wipe and then allowed to air dry. The skin on the prepared area finger was then punctured using a disposable lancet and the first drop of blood wiped away using a clean tissue. Capillary sampling tubes and eppendorf containers were used to collect samples for analysis. Blood lactate analysis was performed once all samples had been collected. Because of the nature of the exercise conditions performed, both total aerobic and total anaerobic energy expenditure estimates were made from mean VO2 across each period and both rest and exercise blood lactate data. This was conducted for each set, rest period, and also the totals across sets, rest periods, and the entire duration of each condition. Estimates were calculated following the approaches described by Scott & Reis (2016) for calculating total energy cost for a given task. It should be noted that for one participant during the leg press condition, and one participant during the recumbent bike condition, the online gas analysis system failed to record data.

Muscle swelling

Muscle swelling was determined from the thickness of the quadriceps (Qt) of the right leg measured using B-mode ultrasound. Participants were instructed to avoid any strenuous physical activity or exercise for at least 3–5 days prior to the acute testing session to avoid acute swelling. Measurements were taken twice at rest prior to the first exercise condition, at 4 min post completion of the first exercise condition, after the additional 20 min of rest between exercise conditions, and at 4 min post completion of the second exercise condition. The participant was placed in a supine position with a pillow placed in the popliteal fossa to relax the upper thigh. The scanning site was identified as the mid-point of the distance from the greater trochanter to the mid-point of the lateral knee joint line. A 7.5 MHz linear array transducer was placed perpendicular to the long axis of the thigh in order to obtain a frozen real-time image of the quadriceps. Measurement of Qt was made in centimetres using the built-in digital callipers measuring the vertical distance from the superficial fat-muscle interface to the underlying femur. Care was taken to ensure adequate contact gel was used and minimal pressure applied to minimise distortion to underlying tissue. All images were be taken by one operator. Intra-operator reliability was determined from using the two baseline measures using standard error of measurement (SEM). The SEM was used to reflect the variation of an individual’s measured values upon repeated testing in order to determine the minimum required observable change in repeated measures to be confident an intervention was responsible. Intra-operator SEM for Qt was 0.36 cm. It should be noted that for one participant it was not possible to obtain a clear enough quadriceps ultrasound image.

Electromyography

Electromyography was measured for the rectus femoris of the right leg of each participant during each exercise period. Electrode placement was made following the recommendations from the Surface Electromyography for the Non-invasive Assessment of Muscles (SENIAM) project (http://www.seniam.org/). Participant’s skin was shaved and then cleaned using an alcohol-free cleansing wipe at the site used for electrode placement. Raw signals were collected at 2,000 Hz and root mean square (RMS) rectified. Peak and mean RMS amplitudes, in addition to integrated electromyography (iEMG; calculated as µV s using the trapezoidal method) were calculated for each set and as an average across each condition. It should be noted that for one participant for both conditions, and one participant during set 4 in the recumbent bike condition, the electromyography system failed to record data

Data analysis

The independent variables were the set or rest period (1, 2, 3, and 4), and the exercise condition (leg press or recumbent cycle ergometer). Dependent variables were peak and mean VO2, peak and mean RER, blood lactate, anaerobic, aerobic, and total energy expenditure, Qt, peak and mean RMS amplitude, iEMG, and both perceived effort and discomfort. Shapiro–Wilk tests were used to examine assumptions of normality of distribution at p > 0.05. For all dependent variables except Qt, within condition comparisons for exercise and rest periods for data meeting assumptions of normality of distribution were made using two way (set/rest period × condition) repeated measures analysis of variance (ANOVA; a Greenhouse-Geisser correction was applied where assumptions of sphericity were found to be violated using Mauchly’s test) and post hoc pairwise comparisons with a Bonferroni correction. Data violating assumptions of normality of distribution were rank transformed for comparison using two way repeated measures ANOVA. Data for Qt met assumptions of normality of distribution and so between conditions comparisons were made using paired samples t-tests. Statistical analysis was performed using IBM SPSS Statistics for Windows (version 22; IBM Corp., Portsmouth, Hampshire, UK) and p < .05 set as the limit for statistical significance.

Results

VO2, RER, blood lactate, and energy expenditure

For the exercise periods two way repeated measures ANOVA revealed no significant effects by ‘set’ for mean VO2 (F(3,15) = 1.633, p = 0.224), peak VO2 (F(3,15) = 2.000, p = 0.157), or aerobic energy expenditure (F(3,15) = 1.873, p = 0.178). Significant effects by ‘set’ were found for mean RER (F(3,15) = 4.908, p = 0.014), peak RER (F(3,15) = 3.650, p = 0.037), blood lactate (F(3,15) = 23.645, p < 0.001), anaerobic energy expenditure (F(3,15) = 20.178, p < 0.001), and total energy expenditure (F(3,15) = 18.114, p < 0.001). There were no significant effects by ‘condition’ for mean VO2 (F(1,5) = 3.825, p = 0.108), peak VO2 (F(1,5) = 1.077, p = 0.347), mean RER (F(1,5) = 2.219, p = 0.197), peak RER (F(1,5) = 1.676, p = 0.252), blood lactate (F(1,5) = 0.130, p = 0.733), aerobic energy expenditure (F(1,5) = 2.552, p = 0.171), anaerobic energy expenditure (F(1,5) = 0.179, p = 0.690), or total energy expenditure (F(1,5) = 0.331, p = 0.590). Significant ‘set x condition’ interactions were found only for mean RER (F(3,15) = 4.944, p = 0.014) where it tended to be higher during the first two sets in the recumbent cycle ergometer condition.

For the rest periods two way repeated measures ANOVA revealed no significant effects by ‘set’ for peak VO2 (F(3,15) = 1.135, p = 0.367). Significant effects by ‘set’ were found for mean VO2 (F(3,15) = 4.940, p = 0.014), mean RER (F(3,15) = 9.742, p = 0.001), peak RER (F(3,15) = 4.616, p = 0.018), and total energy expenditure (F(3,15) = 4.338, p = 0.022). There were no significant effects by ‘condition’ for mean VO2 (F(1,5) = 1.198, p = 0.324), peak VO2 (F(1,5) = 0.616, p = 0.468), mean RER (F(1,5) = 4.171, p = 0.097), peak RER (F(1,5) = 3.496, p = 0.120), or total energy expenditure (F(1,5) = 0.820, p = 0.407). There were no significant ‘set x condition’ interactions for the rest periods.

Descriptive results for VO2, RER, blood lactate, aerobic, anaerobic, and total energy expenditures for each set and condition in addition to where significant post hoc pairwise comparisons were present are shown in Tables 2–5 respectively.

Table 2 Mean (SD) for mean and peak VO2 results for sets, rest periods, and averages.

Variable	Period	Leg press	Recumbent cycle ergometer	
Mean VO2 (ml kg min−1)	Set 1	18.54 (6.35)	26.08 (8.74)	
	Set 2	18.68 (5.29)	26.95 (7.67)	
	Set 3	18.77 (5.91)	28.24 (8.35)	
	Set 4	19.96 (7.57)	28.06 (8.24)	
	Average sets	19.37 (6.50)	27.33 (7.97)	
	Rest 1	15.94 (3.14)	17.35 (2.86)	
	Rest 2	16.15 (3.52)	18.41 (2.73)	
	Rest 3	16.37 (3.36)	18.75 (3.05)	
	Rest 4	16.11 (2.94)	17.67 (2.40)	
	Average rest	16.00 (3.38)	17.72 (2.78)	
	Average whole	17.68 (4.82)	22.52 (5.20)	
Peak VO2 (ml kg min−1)	Set 1	31.87 (7.63)	36.87 (11.25)	
	Set 2	33.54 (12.66)	40.47 (11.74)	
	Set 3	33.01 (10.57)	40.32 (10.42)	
	Set 4	38.33 (14.39)	40.90 (11.73)	
	Average sets	34.75 (11.66)	39.64 (10.79)	
	Rest 1	34.37 (10.90)	35.40 (7.80)	
	Rest 2	37.13 (7.43)	38.14 (7.88)	
	Rest 3	30.85 (14.01)	39.61 (9.62)	
	Rest 4	35.34 (6.60)	39.51 (8.49)	
	Average rest	34.93 (6.81)	37.41 (8.76)	
	Average whole	34.84 (9.45)	38.53 (9.57)	

Table 3 Mean (SD) for mean and peak RER results for sets, rest periods, and averages.

Variable	Period	Leg press	Recumbent cycle ergometer	
Mean RER	Set 1	0.88 (0.05)	1.11 (0.22)	
	Set 2	0.91 (0.06)	1.03 (0.13)	
	Set 3	0.93 (0.06)	0.95 (0,07)	
	Set 4	0.89 (0.05)	0.89 (0.06)	
	Average sets	0.92 (0.02)	0.99 (0.11)	
	Rest 1	1.13 (0.17)	1.36 (0.22)	
	Rest 2	1.09 (0.10)	1.22 (0.12)	
	Rest 3	1.09 (0.08)	1.16 (0.08)	
	Rest 4	1.07 (0.07)	1.12 (0.10)	
	Average rest	1.10 (0.10)	1.22 (0.13)	
	Average whole	1.01 (0.06)	1.11 (0.12)	
Peak RER	Set 1a	1.09 (0.13)	1.33 (0.30)	
	Set 2	1.11 (0.12)	1.32 (0.30)	
	Set 3	1.13 (0.11)	1.20 (0.21)	
	Set 4	1.06 (0.08)	1.14 (0.14)	
	Average Sets	1.12 (0.05)	1.25 (0.22)	
	Rest 1b	1.44 (0.26)	1.69 (0.30)	
	Rest 2	1.40 (0.23)	1.56 (0.19)	
	Rest 3	1.35 (0.17)	1.45 (0.17)	
	Rest 4	1.42 (0.25)	1.38 (0.17)	
	Average rest	1.40 (0.18)	1.53 (0.16)	
	Average whole	1.26 (0.11)	1.39 (0.19)	
Notes.

a Indicates significant effect by ‘set/rest period’ for pairwise comparisons (1 vs 4).

b Indicates significant effect by ‘set/rest period’ for pairwise comparisons (1 vs 3).

Table 4 Mean (SD) for blood lactate results.

Variable	Period	Leg press	Recumbent cycle ergometer	
Blood Lactate (mmol l−1)	Rest	2.74 (1.57)	
	Set 1a	8.63 (4.04)	6.84 (3.26)	
	Set 2b,c	8.55 (4.11)	8.34 (3.39)	
	Set 3	9.40 (3.74)	10.02 (2.81)	
	Set 4	11.97 (3.37)	11.61 (3.87)	
	Average sets	9.68 (3.39)	9.88 (2.88)	
Notes.

a Indicates significant effect by ‘set/rest period’ for pairwise comparisons (1 vs 4).

b Indicates significant effect by ‘set/rest period’ for pairwise comparisons (2 vs 3).

c Indicates significant effect by ‘set/rest period’ for pairwise comparisons (2 vs 4).

Table 5 Mean (SD) for energy expenditure results for sets, rest periods, and totals.

Variable	Period	Leg press	Recumbent cycle ergometer	
Aerobic energy expenditure (Kcal)	Set 1	7.99 (3.04)	11.08 (2.83)	
	Set 2	8.18 (3.08)	11.58 (2.39)	
	Set 3	8.18 (3.06)	12.12 (2.65)	
	Set 4	8.77 (4.23)	12.03 (2.62)	
	Total sets	34.36 (13.83)	46.82 (9.83)	
Anaerobic energy expenditure (Kcal)	Set 1a	7.54 (4.96)	4.99 (4.09)	
	Set 2d	7.54 (5.45)	7.02 (4.87)	
	Set 3	8.71 (5.4)	9.33 (3.54)	
	Set 4	11.95 (4.73)	11.43 (5.46)	
	Total sets	36.30 (21.57)	32.76 (16.14)	
Total energy expenditure (Kcal)	Set 1a	15.53 (7.22)	16.07 (4.85)	
	Set 2c,d	15.72 (8.00)	18.60 (6.02)	
	Set 3	16.89 (7.99)	21.45 (5.27)	
	Set 4	20.71 (8.27)	23.46 (6.55)	
	Total sets	68.85 (30.80)	68.21 (35.46)	
	Rest 1b	25.86 (6.04)	28.65 (3.31)	
	Rest 2	26.41 (7.91)	30.46 (3.56)	
	Rest 3	26.52 (6.09)	31.04 (4.34)	
	Rest 4	26.02 (4.85)	29.28 (3.00)	
	Total rest	105.93 (26.49)	115.57 (9.70)	
	Total whole	176.59 (58.67)	195.15 (21.53)	
Notes.

a Indicates significant effect by ‘set/rest period’ for pairwise comparisons (1 vs 4).

b Indicates significant effect by ‘set/rest period’ for pairwise comparisons (1 vs 3).

c Indicates significant effect by ‘set/rest period’ for pairwise comparisons (2 vs 3).

d Indicates significant effect by ‘set/rest period’ for pairwise comparisons (2 vs 4).

Muscle swelling

Between condition comparisons for using paired samples t- tests revealed no significant difference for Qt measurements taken prior to both conditions (t(7) = 0.836, p = 0.431) and the difference between the measures was within the SEM. Thus, muscle swelling appeared to have returned close to baseline prior to commencement of the second exercise condition. Between condition comparisons for the change in Qt (i.e., difference between pre and post measures for each condition) using paired samples t-tests also revealed no significant difference (t(7) =  − 0.631, p = 0.548). Further, changes exceeded the SEM suggesting a true change to have occurred. Table 6 shows the Qt results.

Table 6 Mean (SD) pre, post and changes for muscle thickness.

Condition	Pre Qt (cm)	Post Qt (cm)	Change Qt (cm)	
Leg press	5.23 (1.00)	5.66 (0.98)	0.43 (0.53)	
Recumbent cycle ergometer	5.01 (1.01)	5.66 (1.04)	0.66 (0.61)	

Electromyography

Two way repeated measures ANOVA on rank transformed data revealed no significant effects by ‘set’ for mean RMS amplitude (F(3,18) = 0.304, p = 0.822), peak RMS amplitude (F(3,18) = 0.070, p = 0.975), or iEMG (F(3,18) = 0.288, p = 0.833). There were no significant effects by ‘condition’ for mean RMS amplitude (F(1,6) = 0.494, p = 0.508), peak RMS amplitude (F(1,6) = 0.475, p = 0.516), or iEMG (F(1,6) = 0.113, p = 0.749). There were also no significant ‘set x condition’ interactions for mean RMS amplitude (F(3,18) = 1.104, p = 0.373), peak RMS amplitude (F(3,18) = 0.752, p = 0.535), or iEMG (F(3,18) = 0.936, p = 0.444). Figure 1 shows electromyography results within and between conditions.

Figure 1 Comparison across sets, and between conditions, for mean RMS amplitude (A), peak RMS amplitude (B), and iEMG (C)—Mean (SD).

Perceptual responses

Two way repeated measures ANOVA revealed significant effects by ‘set’ for both perceived effort (F(3,21) = 12.831, p < 0.001), and discomfort (F(3,21) = 18.945, p < 0.001). Perceived effort significantly differed between set 1 and set 4 (p = 0.008), set 2 and set 3 (p = 0.048), and set 2 and set 4 (p = 0.017). Perceived discomfort significantly differed between set 1 and set 2 (p = 0.009), set 1 and set 3 (p = 0.003), set 1 and set 4 (p = 0.026), and set 2 and set 3 (p = 0.007). There were no significant effects by ‘condition’ for perceived effort (F(1,7) = 0.359, p = 0.568), and discomfort (F(1,7) = 0.281, p = 0.613). There were also no significant ‘set x condition’ interactions for perceived effort (F(3,21) = 0.443, p = 0.725), and discomfort (F(3,21) = 0.763, p = 0.527). Figure 2 shows effort and discomfort results.

Figure 2 Comparison across sets, and between conditions for perceived effort (A), and perceived discomfort (B)—Mean (SD).

Discussion

The aim of the present study was to examine the effects of exercise modality upon acute physiological responses when effort and duration are matched. Participants performed effort and duration matched leg press and recumbent cycle ergometry and VO2, RER, blood lactate, energy expenditure, muscle swelling, and electromyography responses were examined. Overall results indicated that, for the physiological variables examined, responses to resistance training or ‘cardio’ exercise modalities are similar when effort and duration are matched, supporting previous speculation (Fisher & Steele, 2014). This has been reported previously for low effort, duration matched exercise approaches (Vilaca-Alves et al., 2016; Kuznetsov et al., 2011). However, our study would appear to be the first to examine ecologically valid high intensity of effort exercise.

It is typically thought that a high degree of whole body VO2 max is required in order to stimulate improvements in cardiorespiratory fitness (Astrand et al., 2003). Although this is largely unclear (Midgley, McNaughton & Wilkinson, 2006), and the VO2 of any particular exercise bout is primarily dependent upon the muscle mass utilised during the exercise (Astrand et al., 2003; Stromme, Ingier & Meen, 1977). Prior studies examining the VO2 responses to resistance training have in general reported it as less than 50% of whole body VO2 max (Beckham & Earnest, 2000; Bloomer, 2005; Collins et al., 1991; Dudley, 1988; Hurley et al., 1984; Phillips & Ziuratis, 2004; Phillips & Ziuratis, 2003) and as such it has been suggested that it represents a poor stimulus for improving cardiorespiratory fitness (Jung, 2003). However, there are issues with using comparisons of VO2 relative to whole body VO2 max for exercises involving less muscle mass than that used in whole body testing (Steele et al., 2012). VO2 increases in a roughly linear fashion with traditional progressive ‘cardio’ exercise as intensity of effort increases, and indeed, with resistance training modalities there is a disproportionate increase in VO2 cost when exercise is performed to momentary failure compared with when it is stopped prior to failure (Scott & Reis, 2016; Scott & Earnest, 2011). Even with enhanced oxygen delivery, during maximal exercise it would appear that local VO2 of the active musculature is maximal in both resistance training modes (knee extension; Barden et al., 2007) as well as ‘cardio’ exercise modes (cycling; Calbet et al., 2006). The results of this study would appear to support this concept. There was a slightly greater mean VO2 in the recumbent cycle ergometer condition, albeit not significantly different to that during the leg press condition. The fact that this was not significantly different may be the result of a type II statistical error considering the small sample size of the present study. However, it may also be due to the fact that there is continuous concentric muscle actions during recumbent cycling as each leg alternates with each pedal stroke, whereas with the leg press ∼60% of the duration for each repetition was spent under eccentric muscle action. Thus, as concentric muscle actions typically elicit higher VO2 responses compared to eccentric muscle actions (Asmussen, 1953), this is unsurprising. However, we did not find any significant differences between conditions and further, peak VO2 was more similar between the two conditions. Thus, both resistance training and ‘cardio’ exercise modalities when performed in an effort and duration matched manner produce largely similar VO2 responses and thus may offer a similar stimulus for improvements in cardiorespiratory fitness (Steele et al., 2012).

Further, there were similar responses in terms of blood lactate. It has been suggested that accumulation of blood lactate when exercise is performed above the lactate threshold may act as a stimulus for improvement of lactate removal mechanisms (Dalleck et al., 2010; Enoksen, Shalfawi & Tonnessen, 2011), such as an up-regulation of monocarboxylate transporters (Dubochaud et al., 2000). There is a lack of research examining lactate threshold changes in response to resistance training and so far contrasting findings (Bishop et al., 1999; Marcinik et al., 1991). However, high intensity of effort interval based exercise using ‘cardio’ modalities have been shown to improve lactate threshold (Dalleck et al., 2010; Enoksen, Shalfawi & Tonnessen, 2011; Esfarjani & Laursen, 2007). The similar blood lactate response between conditions in the present study suggests resistance training may also offer a similar stimulus to improve lactate removal mechanisms. Indeed the blood lactate levels reported here are similar to that produced by other ‘high intensity interval training’ approaches (Emberts et al., 2013) as well as those reported in other studies of high effort resistance training (Hurley et al., 1984; Charro et al., 2010; Denton & Cronin, 2006; Gentil, Oliveira & Bottaro, 2006; Tesch, Colliander & Kaiser, 1986).

Aerobic, anaerobic, and total energy expenditures were also similar between both conditions. In other investigation of work and duration matched running and cycling, despite slight differences in VO2 as seen here, there were also similar energy costs (Scott et al., 2006). This is also the case with different work matched resistance training approaches (Scott, 2006; Aniceto et al., 2013). There were no significant between condition differences for either aerobic or anaerobic energy expenditures, although it did appear that anaerobic energy expenditure during the exercise periods exceeded aerobic energy expenditure for the leg press conditions, and this was reversed in the recumbent cycle ergometer condition. Further, aerobic energy costs remained similar across sets between conditions, yet blood lactate, and thus anaerobic energy costs, appeared to increase across sets for both conditions presumably as a result of cumulative fatigue and/or adjustment of load/resistance resulting in the later sets having a closer proximity to failure. This also likely influenced total energy costs across sets which increased. Indeed, fatiguing exercise resulting in failure appears to result in greater anaerobic and total energy costs compared with non-fatiguing exercise ending prior to failure (Scott & Earnest, 2011). Total energy expenditure was higher for rest periods as compared with exercise as has been shown for brief, intermittent higher intensity of effort exercise in a range of modalities (Scott & Reis, 2016; Scott & Earnest, 2011; Scott, 2006; Scott et al., 2006). However, rate of energy expenditure was higher during exercise (∼17 Kcal min−1) compared with during rest (∼9 Kcal min−1). Although our RER data is unclear as to substrate use during the conditions, and was likely influenced by hyperventilation due to the high effort nature of the exercise, the similar total energy expenditures of both modalities suggests also that both may also offer similar benefits in terms of body mass loss efforts. Thus, effort and duration of exercise may be the more influential moderators of this and indeed other work suggests resistance and aerobic training appear to offer similar benefits in terms of body mass loss (Villareal et al., 2017).

In addition to the potentially similar stimulus and benefits to cardiorespiratory fitness (Steele et al., 2012; Ozaki et al., 2013a), the present results also appear to support a similar acute stimulus for strength and hypertrophic adaptations. It is typically thought that mechanical tension from high external loads is a primary stimulus for strength and hypertrophic adaptation (Schoenfeld, 2010). However, accumulation of metabolites such as lactate may result in so called ‘metabolic stress’ which may also impact on factors such as cellular swelling and motor unit recruitment to potentially produce adaptations even without high mechanical tension from external loads (Counts et al., 2016; Gentil et al., 2017; Schoenfeld, 2013). This has mostly been considered with respect to why low load resistance training may produce similar adaptations to high load resistance training (Schoenfeld, 2013; De Freitas et al., 2017); however, it has also been argued it may apply to other modalities typically performed with low external loads (i.e., ‘cardio’ exercise modes; Ozaki et al., 2016). As already noted, blood lactate response in both conditions was similar indicating similar levels of metabolite accumulation, and subsequently, similar changes in Qt occurred indicating similar degrees of muscular swelling. Although prior studies have reported that both ‘cardio’ exercise and resistance training performed to high intensities of effort can induce increases in muscle water content and produce acute muscular swelling (Ozaki et al., 2013b; Giessing et al., 2016; Ribiero et al., 2014; Loenneke et al., 2016; Mora-Rodriguez et al., 2016), this appears to be the first to directly compare these responses between modalities when effort and duration matched. Cellular swelling has also been argued to be a trigger associated with the proliferation of satellite cells and thus a contributor to the hypertrophic response to exercise (Schoenfeld, 2013; De Freitas et al., 2017). In fact, other recent work (Nederveen et al., 2015) comparing high effort resistance training (leg press and knee extension using 95% 10RM for 2 sets of 10 repetitions and a final set to momentary failure) and ‘high intensity interval training’ using a ‘cardio’ modality (cycling for 10 sets of 60 s at 90–95% of VO2 peak with 60 s recovery at 50 W between sets), has reported that satellite cell responses in both of these modes were largely similar and higher than continuous moderate effort cycling (30 min at 55–60% of VO2 peak) As such, both resistance training and ‘cardio’ exercise modalities may offer a similar stimulus for hypertrophic adaptations when performed in an effort matched manner in a similar way to high and low load resistance training (Morton et al., 2016).

As noted, strength may be optimised through training using high external loads (Schoenfeld et al., 2017). However, this is thought to be primarily due to specificity in motor learning and mostly occurs when strength testing closely reflects the training performed (Fisher, Steele & Smith, 2017; Dankel et al., 2017; Mattocks et al., 2017). When training and testing are dissimilar (e.g., dynamic training performed but isometric testing conducted) both high and low external loads produce similar changes in strength (Fisher, Ironside & Steele, 2017). This is thought to be due to Henneman’s size principle, and recent modelling studies suggest that motor unit recruitment should be similar between muscular actions when they are performed to a near maximal effort (Potvin & Fuglevand, 2017). Motor unit recruitment patterns are difficult to discern specifically from amplitude based surface EMG analyses (Fisher, Steele & Smith, 2017; Vigotsky et al., 2017; Enoka & Duchateau, 2015). Indeed, if tasks being compared differ considerably in effort and/or durations, motor unit recruitment patterns may differ (e.g., synchronous vs sequential) such that, even though similar numbers of total motor units are recruited, EMG amplitudes may differ (Fisher, Steele & Smith, 2017; Vigotsky et al., 2017). However, in the present study attempts were made to approximately match the efforts and durations of each exercise condition. As such, amplitude based EMG analyses were compared between conditions to give an indication of motor unit recruitment and all appeared to be similar between conditions. These results, in agreement with others (Kuznetsov et al., 2011) suggest that, when effort and duration matched, both resistance training and aerobic exercise modalities using similar muscle groups likely result in similar motor unit recruitment despite differences in external loads.

The exercise conditions in this study were designed with an attempt to maximise perceived effort per set and match it between conditions. However, our data suggested that on average perceived effort was high, but not maximal. Typically it increased across sets presumably either as a function of accumulated fatigue, and/or adjustments made to load/resistance, resulting in later sets being performed with closer proximity to failure. However, there were no differences between conditions suggesting that efforts were successfully matched between both the leg press and recumbent cycle ergometer. Discomfort scales were also used in order to get participants to disentangle their perceptions of effort from associated physical discomfort as it is common for people to anchor one upon the other with traditional rating of perceived exertion scales (Steele et al., 2017c). Discomfort also increased across sets, yet without differences between conditions. Discomfort has been suggested to be more closely related to afferent feedback (Marcora, 2009) and the similar increases seen here between both conditions may be a result of similar stimulation of group III and IV fibres providing afferent feedback about the metabolic and mechanical conditions induced by the exercise. Indeed as noted, the physiological responses examined were largely similar between conditions including blood lactate increases, muscle swelling, and EMG amplitudes. In studies of resistance training where effort has been matched yet other variables manipulated, for example load or the use of advanced training techniques, discomfort can differ (Fisher, Ironside & Steele, 2017; Fisher, Farrow & Steele, 2017). Yet these results suggest that when effort and duration are matched the discomfort experienced is also similar between exercise modalities utilising similar muscle groups.

The results of this study suggest that, at least when considering the variables measured (i.e., VO2, RER, blood lactate, energy expenditure, muscle swelling, and electromyography), the physiological stimulus provided by both resistance training (leg press), and ‘cardio’ exercise (recumbent cycle ergometry), when effort and duration matched may be largely similar. As such, similar chronic physiological adaptations may possibly occur from either modality when performed with a similar effort and duration. However, there is currently a lack of research examining the chronic adaptations to appropriately matched resistance training and ‘cardio’ exercise modalities. Studies directly comparing resistance training and ‘cardio’ training modalities upon these chronic adaptations contrast in their findings (Farup et al., 2012; Goldberg, Elliot & Kuehl, 1994; Hepple et al., 1997; Jubrias et al., 2001; Messier & Dill, 1985; Sawczyn et al., 2015; Wilkinson et al., 2008) and this may be due to the fact that effort and duration have not been matched. Resistance training is typically performed with a relatively high effort (though not always) and with varying durations, and ‘cardio’ exercise is typically performed with a low to moderate effort and high durations. Yet a recent study has suggested that, when performed in an effort and duration matched manner, eight weeks additional resistance training mode (squats and deadlifts), or ‘cardio’ exercise mode (upright cycle ergometry) ‘high intensity interval training’ produces largely similar improvements in cardiorespiratory fitness and knee extension strength in powerlifting and strongman athletes (Androulakis-Korakkakis et al., 2017). Considering the highly trained nature of the population in that study, similar results might occur in an untrained population. Indeed, another study examining 12 weeks of effort and duration matched ‘cardio’ (cycle ergometer ‘high intensity interval training’) and resistance training (full body resistance training including biceps curls, knee extension, shoulder press, and upright rows) also showed that a range of outcomes were similar in insulin resistant women (Álvarez et al., 2017). Future work should thus look to further examine effort and duration matched interventions, such as that used here, upon chronic physiological adaptations in cardiorespiratory fitness, strength, and hypertrophy across a range of populations. Such work should also be careful to avoid the confounding influence of specificity of training and testing and use outcomes that will not bias towards one condition vs the other. For example, training using a recumbent cycle ergometer and leg press, and testing for VO2 max using an upright cycle ergometer or treadmill, and testing for strength using isometric knee extension.

The limitations of the present study should be acknowledged. The sample size was relatively small for this study. Although we utilised a within session randomised crossover design in order to increase the statistical power of our comparisons, it may be that some of the variables examined were not found to be statistically significantly different as a result of a type II error. Although, it could contrastingly be argued that, as a wide range of different physiological responses were examined, it might have been expected that the chances of a type I error might have been increased. Future work should perhaps look to replicate our study design with a larger sample to clarify if indeed the physiological responses to differing exercise modalities are similar when effort and duration matched. It should also be noted that all of the variables measured were examined during and immediately after the exercise periods. Measures such as hormonal concentrations or biochemical signalling pathways relevant to understanding the stimulus provided by an exercise bout were not examined yet may differ dependent upon modality. However, similarly to the responses examined here, previous research has generally not controlled appropriately for both effort and duration when comparing signalling responses across modalities (Wilkinson et al., 2008). As such, future work may also look to examine other acute responses, during, and for longer periods after, the exercise bouts (i.e., up to 24–48 h post exercise).

Conclusions

The present study shows that, when both effort and duration are matched, resistance training (leg press) and ‘cardio’ exercise (recumbent cycle ergometry) may produce largely similar responses in VO2, RER, blood lactate, energy expenditure, muscle swelling, and electromyography. It therefore seems reasonable to suggest that both may offer a similar stimulus to produce chronic physiological adaptations in outcomes such as cardiorespiratory fitness, strength, and hypertrophy. Although there is limited research comparing the chronic adaptations from such matched exercise modalities, the implications of such findings might be significant. If modality has little impact upon changes in physiological fitness outcomes known to be strong predictors of health and longevity then there may be potential for public health messages about physical activity and exercise to be widened to increase participation. Persons may be able to select their exercise mode based upon personal preferences and/or circumstances, and instead focus upon a key message to engage in such activities with a high intensity of effort. Future work should look to both replicate the study conducted here with respect to the same, and additional physiological measures, and rigorously test the comparative efficacy of effort and duration matched exercise of differing modalities with respect to chronic improvements in physiological fitness.

Supplemental Information

Supplemental Information 1 VO2, Lactate, EE data

Click here for additional data file.

Supplemental Information 2 Acute UT data

Click here for additional data file.

Supplemental Information 3 EMG data

Click here for additional data file.

Supplemental Information 4 Perceptual data

Click here for additional data file.

Additional Information and Declarations

Competing Interests

Author Contributions

Human Ethics

Data Availability

The authors declare there are no competing interests.

James Steele conceived and designed the experiments, performed the experiments, analyzed the data, prepared figures and/or tables, authored or reviewed drafts of the paper, approved the final draft.

Andrew Butler performed the experiments, authored or reviewed drafts of the paper, approved the final draft.

Zoe Comerford, Jason Dyer, Nathan Lloyd and Joshua Ward performed the experiments, analyzed the data, authored or reviewed drafts of the paper, approved the final draft.

James Fisher, Paulo Gentil, Christopher Scott and Hayao Ozaki conceived and designed the experiments, authored or reviewed drafts of the paper, approved the final draft.

The following information was supplied relating to ethical approvals (i.e., approving body and any reference numbers):

The study was approved by the Health, Exercise, and Sport Science ethics committee at Southampton Solent University.

The following information was supplied regarding data availability:

The raw data is available as a Supplemental File.

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
