# Peer review of "Similar acute physiological responses from effort and duration matched leg press and recumbent cycling tasks"

_PeerJ, doi:10.7717/peerj.4403_

## Round 0.1 · original submission · Major Revisions

· Academic Editor

Major Revisions

Dear Authors,

Although two out of three reviewers give a relative positive evaluation of your manuscript, I tend to be more in agreement with the first reviewer.

Please, provide us with a solid point-by-point response to further consider your manuscript for publication.

Kind regards.

Reviewer 1 ·

Basic reporting

The authors mentioned that " The adaptations in cardiorespiratory fitness typically thought to occur as a result of modalities of exercise such as cycling also have been reported to occur as a result of resistance training assuming intensity of effort is sufficiently 96 high (i.e. to momentary failure; Steele et al., 2012) irrespective of the manipulation of other variables (i.e. load, set volume, rest periods, and frequency; Ozaki et al., 2013a).” However, it is quite tendencious since there are several studies which showed that cardiorespiratory adaptations are minimal following resistance training, specially when compared to endurance training.

The authors also mentioned that " Conversely, the adaptations in strength and hypertrophy thought typically to occur from resistance training modalities of exercise have been found to occur as a result of aerobic modalities (Konopka and Harber, 2014), particularly if performed with a high effort (i.e. combined with 100 blood flow restriction, or with close proximity to failure; de Oliviera et al., 2016; Lundberg et al., 2013; Ozaki et al., 2015; Ozaki et al., 2013b). However, increases in maximal strength and muscle mass have been shown only following specific endurance modalities, such as rowing, cycling and, sprint running. Therefore, the authors’ statement is an oversimplification of the information.

Again, the authors mentioned that " it is thought that, similarly to ‘cardio’ modalities, resistance training to a high intensity of effort results in maximal aerobic and anaerobic metabolism, and as such local working muscle oxygen utilisation (VO2 ) and lactate production (Steele et al., 2012).” Ok. there is a study. But, how about the others several studies showing a quite low oxygen uptake during resistance training? The authors should also mention the opposite results in order to bring the state of the art of the subject.

In general, the “Introduction" section is very confuse and the research problem is not sound. It is not clear why the authors are comparing something that have been extensively compared in the literature and induces quite different adaptations following a training period.

The mention of the Henneman’s size principle and its association with the resistance training performed at high or low loads performed to a near maximal effort is quite confuse and speculative.

Experimental design

Methods
The sample size in very small.

The Experimental design does not support the authors' conclusion.

Validity of the findings

Discussion and conclusion

The authors discussed and concluded that the “physiological responses” were similar, but actually, if we take a look on the results, there are quite different values in some variables that probably were not statistically different because of the insufficient sample size. In addition, the Henneman’s size principle is equivocally mentioned in the Discussion section.
The statement "It therefore seems reasonable to suggest that both may offer a similar stimulus to produce chronic physiological adaptations in outcomes such as cardiorespiratory fitness, strength, and hypertrophy.” is absolutely not supported by the results. What were the parameters investigated that could suggest that strength and hypertrophy are similarly stimulated following endurance and strength training? I did not find anyone to state such strong affirmation.

Comments for the author

The purpose of this study was "to examine the effects of training utilising traditional resistance training (leg press)or ‘cardio’ exercise (recumbent cycle ergometry) modalities whilst attempting to approximately control for effort and duration within ecologically valid protocols, upon acute physiological responses (VO2 , respiratory exchange ratio [RER], blood lactate, energy expenditure, muscle swelling, and electromyography). Along with the absence of originality, there are several concerns regarding the theory presented in the “Introduction” section to support the research problem; insufficient sample size to assess such parameters, specially electromyography; dissimilar physiological responses between conditions, although non-significant different (probably due to the sample size), in some important parameters, that are discussed as similar; among others. Please find below the specific comments.

Reviewer 2 ·

Basic reporting

no comment

Experimental design

no comment

Validity of the findings

no comment

Comments for the author

The present study aimed to examine the effects of training utilising traditional resistance training (leg press) or ‘cardio’ exercise (recumbent cycle ergometry) modalities whilst attempting to approximately control for effort and duration within ecologically valid protocols, upon acute physiological responses. This manuscript is well written. Congratulations. Future study should really focus on examining other acute responses, during and for longer periods after the exercise bouts.

Reviewer 3 ·

Basic reporting

Title: Similar acute physiological responses from effort and duration matched leg press and recumbent cycling tasks

Steele et al measured the effects of exercise utilising traditional resistance training (leg press) or ‘cardio’ 32 exercise (recumbent cycle ergometry) modalities upon acute physiological responses.

Nine healthy males underwent 33 a within session randomised crossover design. Conditions were approximately matched for effort and duration (leg press: 4 x 12RM, recumbent cycle ergometer: 4 x 60 second sprints). Measurements included VO2, respiratory exchange ratio (RER), 36 blood lactate, energy expenditure, muscle swelling, and electromyography. Perceived effort was similar between 37 conditions and thus both were well matched with respect to effort. The present study shows that, when both effort and duration are matched, resistance training and ‘cardio’ exercise produce largely similar physiological responses.

Experimental design

Minor concerns
INTRODUCTION:
-Line 64 use of capital letter (non-necessary)
-Line 66 (Lee et al, 2011) Are there some reference more updated?
-Line 75 There is no reference to this paragraph.
-Line 79….Whe authors says “high effort”,..can the authors explain about what effort? I mean high 1RMs…or looking for fatigue of muscles into a specific time usin resistant training?
-resistant training can include these abvreviations (RT)
-The authors could include some other similar effects of RT than HIT…over body composition: “Álvarez, C., Ramírez-Campillo, R., Ramírez-Vélez, R., & Izquierdo, M. (2017). Effects and prevalence of nonresponders after 12 weeks of high-intensity interval or resistance training in women with insulin resistance: a randomized trial. Journal of Applied Physiology, 122(4), 985-996”. This in order to compare than RT can be similar than HIT..or other exercise modalities at body composition….including performance.
-Line 181…Says Stature..can be changed by “Height”.

Validity of the findings

The literature is appropriate, and please to this section comments to review my general main and minor comments.

Comments for the author

Main concerns.
-How can the author extrapolate or equally 12RM to 60 seconds? I sure that readers can be more clear if the author can explain a bit more about this.

-Although I can understand as scientific the methodology, I have appreciate that the CONCLUSION is to much strong in this “, when both effort and duration are matched, resistance training and ‘cardio’ exercise produce largely similar physiological responses”, where I recommend to be more specific/or to add some comment about in what physiological terms are these similar physiological terms.

-The author could include in the conclusion section at what specific physiological concern (VO2, RER, blood lactate, ee, muscle swelling, or electromyography)…are these equal physiological responses.

-Line 88. To reconsider re-organize into the others paragraph this last.

-I strongly recommend to reduce the INTRODUCTION section.

---

## Round 0.2 · Minor Revisions

· Academic Editor

Minor Revisions

I am broadly satisfied with the sophistication that authors have bring into the new version of the manuscript. However, some issues preclude me to make a decision regarding its acceptability. Author`s should address some issues before further consideration.

Specific comments

I am not sure about the adequacy of foot notes. Please remove and place the information-citation in the text.

Also, please carefully review the whole manuscript for typos (e.g.: Line 79, “resistance bands etc.)”, it seems that a coma is missing; line 83; etc.).

Lines 152-156: Please provide the hypothesis.

Line 265: Please avoid start a sentence with an acronym (or number). Check through the whole manuscript please (e.g., line 269).

Table 2 is a little bit long. Can authors provide an alternative configuration?

Reviewer 3 ·

Basic reporting

I found improved this section after my first comments.

Experimental design

I found improved this section after my first comments.

Validity of the findings

I found improved this section after my first comments.

Comments for the author

I found improved this section after my first comments.

---

## Round 0.3 · accepted · Accept

· Academic Editor

Accept

I am pleased to inform you of the official acceptance of your manuscript for publication in PeerJ.